# Entanglement Swapping and Swapped Entanglement

**DOI:** 10.3390/e25030415

**Published:** 2023-02-25

**Authors:** Sultan M. Zangi, Chitra Shukla, Atta ur Rahman, Bo Zheng

**Affiliations:** 1School of Physics and Astronomy and Yunnan Key Laboratory for Quantum Information, Yunnan University, Kunming 650500, China; 2Shenzhen Institute for Quantum Science and Engineering and Department of Physics, Southern University of Science and Technology, Shenzhen 518055, China; 3School of Physics, University of Chinese Academy of Sciences, Yuquan Road 19A, Beijing 100049, China; 4Collaborative Innovation Center of Advanced Microstructures, Nanjing University, Nanjing 210093, China

**Keywords:** entanglement, swapping, negativity, concurrence

## Abstract

Entanglement swapping is gaining widespread attention due to its application in entanglement distribution among different parts of quantum appliances. We investigate the entanglement swapping for pure and noisy systems, and argue different entanglement quantifiers for quantum states. We explore the relationship between the entanglement of initial states and the average entanglement of final states in terms of concurrence and negativity. We find that if initial quantum states are maximally entangled and we make measurements in the Bell basis, then average concurrence and average negativity of final states give similar results. In this case, we simply obtain the average concurrence (average negativity) of the final states by taking the product of concurrences (negativities) of the initial states. However, the measurement in non-maximally entangled basis during entanglement swapping degrades the average swapped entanglement. Further, the product of the entanglement of the initial mixed states provides an upper bound to the average swapped entanglement of final states obtained after entanglement swapping. The negativity work well for weak entangled noisy states but concurrence gives better results for relatively strong entanglement regimes. We also discuss how successfully the output state can be used as a channel for the teleportation of an unknown qubit.

## 1. Introduction

Schrödinger observed the existence of a unique correlation, called entanglement, in some quantum states of two or more systems at the dawn of quantum mechanics [1]. Now, it is well understood that entanglement is a purely quantum phenomena [2,3]. The creation, modification, control, and practical application of this quantum correlation have become significant research fields [4,5]. Specially, the ability to distribute entanglement between distant systems has the potential to be used in the development of novel quantum information protocols [6,7].

The entanglement swapping is a mechanism for distributing the entanglement correlation between distant systems. There has been a great deal of theoretical and experimental research completed on entanglement swapping [8,9,10,11,12,13]. It also helps us to connect many separable nodes for long-distance communication in a quantum network [14,15]. Specifically, entanglement swapping is a protocol by which quantum systems that have never interacted in the past can become entangled [6,16]. The nomenclature “entanglement swapping” describes the transfer of entanglement from a priori entangled systems to a priori separable systems [17]. It is a very useful tool for entanglement purification [18], teleportation [19], and plays an important role in quantum computing and quantum cryptography [4,20]. We can also use entanglement swapping for the creation of multipartite entangled states from bipartite entanglement [21].

Let us describe the phenomenon of entanglement swapping. Suppose two entangled particles (A,B) are shared between Alice and Bob. Similarly Cara and Danny also share another entangled pair of particles (C,D). Initially there is no entanglement between Alice’s and Danny’s particles (A,D), shown in Figure 1a. If Bob and Cara who are situated in the same laboratory make measurement in a suitable basis on the pair (B,C) and classically communicate the outcome with distant partners then Alice’s and Danny’s particles who are at very large distance become entangled as shone in Figure 1b. This entanglement swapping protocol can be generalized in different ways: by modifying the initial states, or by modifying the measurement performed by Bob and Cara, or by extending the number of parties [21,22].

An ever-increasing body of literature shows that the entanglement swapping and purification of quantum systems need specific protocols. Smaller changes may bring huge changes to the output state because of the relative sensitivity of the operation and quantum systems. For this reason, several previous studies suggest entanglement swapping of initial states into maximally entangled states. For example, in Ref. [23], the authors provided a scheme of entanglement swapping of initial states into biqubit maximally entangled states when influenced by an amplitude damping channel. The concurrence of the measuring basis for entanglement swapping caused a two-fold entanglement matching effect has been witnessed in Ref. [24]. The authors of Ref. [25] showed relationship between the ranks of the initial states and the rank of the final state after swapping, as well as that concurrence of partially mixed states remains conserved when is swapped with the Bell state. Our research discusses the complement to the earlier findings in the cited reference. Concurrence-based weak entanglement regimes are not detected in the mixed state. It is interesting to note that negativity is still more robust than concurrence at the weak entanglement regimes of the mixed state when we revisit this using negativity in our work. Recently, Ref. [26] investigated that hyper-entangled states produce deterministic entanglement swapping while considering projections of biqubit systems on symmetric, and iso-entangled states. It is found that biqubit entanglement generated through entanglement swapping, will depart from a Bell-type inequality even for visibilities smaller than 50% [9]. From the above literature, we found that entanglement swapping has been previously considered using different procedures, and various important results have been achieved. This research work constitutes a relatively more generalized entanglement swapping protocol that covers the maximum possible cases of swapping. We consider pure, mixed, and noisy systems for entanglement swapping. Ordering the quantum systems with respect to different entanglement quantifiers is also one of the hot issues of quantum information [27,28,29,30]. Here, we also investigate the ordering of final states via concurrence [31] and negativity [32,33]. We compare the results of concurrence and negativity and find their association under different circumstances. We specifically wish to investigate the relationship between the average concurrence and average negativity of the output states and the concurrences and negativities of the input states. In case of pure states, it is discovered that the average concurrence (average negativity) of the final state is simply the sum of the concurrences (negativities) of the states used to perform the entanglement swap. Then, for noisy qubits, concurrence (negativity) and entanglement swapping are discussed. We observe that week entangled final noisy states are only detected by negativity but in the relatively higher entangled noisy state domains, concurrence remains more favorable. Moreover, we explore the application of the final state as a channel for the teleportation of an unknown qubit. We also investigate the fidelity of teleported qubit with the initial unknown qubit. One of the intentions of our work is to provide a theoretical scheme that can easily be used in experiments containing swapping protocol. In this case, we demonstrated an easily applicable entanglement swapping design compared to the previous studies [9,23,24,25,26].

The article is organized as follows. Section 2 describes the details of the entanglement swapping scenario for non-noisy systems and the application of swapped entanglement. Next, in Section 3, we demonstrate the entanglement swapping among noisy qubits and teleportation using a noisy quantum channel. Finally, we conclude with a short discussion in Section 4.

## 2. Entanglement Swapping among Qubit Systems

We consider two pairs of qubits for entanglement swapping. The entangled qubits *A* and *B* make the first pair and the second pair consist of entangled qubits *C* and *D* (the qubits’ names are chosen such that they make the initial of Alice, Bob, Cara, and Danny, shown in Figure 1). In terms of Schmidt decomposition, these subsystems can be written as
(1)$$\begin{array}{cc} {|\phi \rangle }_{AB} & =\sqrt{p_{0}}{|00\rangle }_{AB}+\sqrt{p_{1}}{|11\rangle }_{AB},\\ {|\phi \rangle }_{CD} & =\sqrt{p_{0}^{\prime }}{|00\rangle }_{CD}+\sqrt{p_{1}^{\prime }}{|11\rangle }_{CD}\;. \end{array}$$The concurrence and negativity of these systems are $C\left({|\phi \rangle }_{AB}\right)=2\sqrt{p_{0}p_{1}}$, $C\left({|\phi \rangle }_{CD}\right)=2\sqrt{p_{0}^{\prime }p_{1}^{\prime }}$ and $N\left({|\phi \rangle }_{AB}\right)=2\sqrt{p_{0}}\sqrt{p_{1}}$, $N\left({|\phi \rangle }_{CD}\right)=2\sqrt{p_{0}^{\prime }}\sqrt{p_{1}^{\prime }}$. It means for these bi-dimensional systems concurrence and negativity produce similar results. The initial state of our four-qubit system is $|\Phi \rangle ={|\phi \rangle }_{AB}⊗{|\phi \rangle }_{CD}$ and after rearrangement of the qubits *A* and *D* together and qubits *B* and *C* together we can write
(2)$$\begin{array}{c} |\Phi \rangle =\sqrt{p_{0}p_{0}^{\prime }}{|00\rangle }_{AD}{|00\rangle }_{BC}+\sqrt{p_{0}p_{1}^{\prime }}{|01\rangle }_{AD}{|01\rangle }_{BC}+\\ \sqrt{p_{1}p_{0}^{\prime }}{|10\rangle }_{AD}{|10\rangle }_{BC}+\sqrt{p_{1}p_{1}^{\prime }}{|11\rangle }_{AD}{|11\rangle }_{BC}\;. \end{array}$$In order to do measurements over BC qubits, we also need to define a set of four orthonormal basis [24]
(3)$$\begin{array}{cc} {|{\tilde{\Phi }}_{+}\rangle }_{BC} & =\alpha_{0}{|00\rangle }_{BC}+\beta_{0}{|11\rangle }_{BC},\\ {|{\tilde{\Phi }}_{-}\rangle }_{BC} & =\beta_{0}^{*}{|00\rangle }_{BC}-\alpha_{0}^{*}{|11\rangle }_{BC},\\ {|{\tilde{\Psi }}_{+}\rangle }_{BC} & =\alpha_{1}{|01\rangle }_{BC}+\beta_{1}{|10\rangle }_{BC},\\ {|{\tilde{\Psi }}_{-}\rangle }_{BC} & =\beta_{1}^{*}{|01\rangle }_{BC}-\alpha_{1}^{*}{|10\rangle }_{BC}\;, \end{array}$$
where $\alpha_{i}$ and $\beta_{i}$ are unknown coefficients and for normalization ${\left|\alpha_{i}\right|}^{2}+{\left|\beta_{i}\right|}^{2}=1$ for i∈{0,1}. Conversely, we have ${|00\rangle }_{BC}=\alpha_{0}^{*}{|{\tilde{\Phi }}_{+}\rangle }_{BC}+\beta_{0}{|{\tilde{\Phi }}_{-}\rangle }_{BC}$ and, likewise, we can find expressions for ${|01\rangle }_{BC},{|10\rangle }_{BC},{|11\rangle }_{BC}$. Now by using these expressions we can write Equation (Equation 2) as
(4)$$\begin{array}{c} |\Phi \rangle =\sqrt{p_{{\tilde{\Phi }}_{+}}}{|{\ddot{\Phi }}_{+}\rangle }_{AD}{|{\tilde{\Phi }}_{+}\rangle }_{BC}+\sqrt{p_{{\tilde{\Phi }}_{-}}}{|{\ddot{\Phi }}_{-}\rangle }_{AD}{|{\tilde{\Phi }}_{-}\rangle }_{BC}\\ +\sqrt{p_{{\tilde{\Psi }}_{+}}}{|{\ddot{\Psi }}_{+}\rangle }_{AD}{|{\tilde{\Psi }}_{+}\rangle }_{BC}+\sqrt{p_{{\tilde{\Psi }}_{-}}}{|{\ddot{\Psi }}_{-}\rangle }_{AD}{|{\tilde{\Psi }}_{-}\rangle }_{BC}\;, \end{array}$$
where ${|{\ddot{\Phi }}_{+}\rangle }_{AD}=\left(\sqrt{p_{0}p_{0}^{\prime }}\alpha_{0}^{*}{|0\rangle }_{A}{|0\rangle }_{D}+\sqrt{p_{1}p_{1}^{\prime }}\beta_{0}^{*}{|1\rangle }_{A}{|1\rangle }_{D}\right)/\sqrt{p_{{\tilde{\Phi }}_{+}}}$ with probability $p_{{\tilde{\Phi }}_{+}}=p_{0}p_{0}^{\prime }{\left|\alpha_{0}\right|}^{2}+p_{1}p_{1}^{\prime }{\left|\beta_{0}\right|}^{2}$, similarly ${|{\ddot{\Phi }}_{-}\rangle }_{AD}$, ${|{\ddot{\Psi }}_{+}\rangle }_{AD}$, and ${|{\ddot{\Psi }}_{-}\rangle }_{AD}$ can be obtained.

We observe in Equation (Equation 4) that the state of qubits *A* and *D* is similar to the state of the basis BC. After measurements in the basis BC, Alice and Danny’s qubits *A*, *D* which are initially separable, become entangled in one of the four possible forms. We can compute the average concurrence for the final state as
(5)$$\begin{array}{cc} C_{av} & =p_{{\tilde{\Phi }}_{+}}C_{{\ddot{\Phi }}_{+}}+p_{{\tilde{\Phi }}_{-}}C_{{\ddot{\Phi }}_{-}}+p_{{\tilde{\Psi }}_{+}}C_{{\ddot{\Psi }}_{+}}+p_{{\tilde{\Psi }}_{-}}C_{{\ddot{\Psi }}_{-}}\;,\\ \; & =4\sqrt{p_{0}p_{0}^{\prime }p_{1}p_{1}^{\prime }}\left(\left|\alpha_{0}\beta_{0}\right|+\left|\alpha_{1}\beta_{1}\right|\right)\;. \end{array}$$

The Bell states are maximally entangled biqubit states. The states in Equation (Equation 3) transform into maximally entangled Bell states if we take $\alpha_{i}=\beta_{i}=\frac{1}{\sqrt{2}}$ for i∈{0,1}. The measurement in maximally entangled Bell basis return maximally entangled *A*, *D* qubits states with the average concurrence
(6)$$\begin{array}{cc} C_{av} & =4\sqrt{p_{0}p_{0}^{\prime }p_{1}p_{1}^{\prime }}\;\\ \; & =C_{AB}C_{CD}\;. \end{array}$$Similarly, the average negativity of the qubits *A* and *D* states when the measurement is completed in Bell basis takes the form
(7)$$\begin{array}{cc} N_{av} & =4\sqrt{p_{0}p_{0}^{\prime }p_{1}p_{1}^{\prime }}\;\\ \; & =N_{AB}N_{CD}\;. \end{array}$$We obtain from Equations (Equation 6) and (Equation 7) that if initial quantum states are maximally entangled and we make measurements in the Bell basis, then average concurrence and average negativity are equivalent. We simply obtain the average concurrence (average negativity) by taking the product of concurrences (negativities) of the initial states. In addition, Equation (Equation 5) shows that measurement in non-maximally entangled basis during entanglement swapping degrades the average swapped entanglement.

Now we extend entanglement swapping between two pairs of qubits to three pairs of qubits. We take three pairs of entangled qubits and make a measurement in the GHZ basis and analyze the outcome state. If the three pairs of qubits in Schmidt form are ${|\phi \rangle }_{AB}=\sqrt{\lambda_{0}}{|00\rangle }_{AB}+\sqrt{\lambda_{1}}{|11\rangle }_{AB})$, ${|\phi \rangle }_{CD}=\sqrt{\mu_{0}}{|00\rangle }_{CD}+\sqrt{\mu_{1}}{|11\rangle }_{CD}$ and ${|\phi \rangle }_{EF}=\sqrt{\nu_{0}}{|00\rangle }_{EF}+\sqrt{\nu_{1}}{|11\rangle }_{EF}$ then the six-qubit system can be written as
(8)$$|\Phi^{\prime }\rangle ={|\phi \rangle }_{AB}⊗{|\phi \rangle }_{CD}⊗{|\phi \rangle }_{EF}.$$Let us make measurements on the *B*, *D*, and *F* qubits. For this purpose, we can define the triqubit GHZ basis as [34]
(9)$$\begin{array}{cc} |G_{0,1}\rangle & =\frac{1}{\sqrt{2}}(|000\rangle \pm |111\rangle )\;,\\ |G_{2,3}\rangle & =\frac{1}{\sqrt{2}}(|001\rangle \pm |110\rangle )\;,\\ |G_{4,5}\rangle & =\frac{1}{\sqrt{2}}(|010\rangle \pm |101\rangle )\;,\\ |G_{6,7}\rangle & =\frac{1}{\sqrt{2}}(|011\rangle \pm |100\rangle )\;. \end{array}$$Here, the − sign applies to states with odd indices. Now we can write Equation (Equation 8) in terms of GHZ basis as
(10)$$\begin{array}{cc} |\Phi^{\prime }\rangle & =\frac{1}{\sqrt{2}}\left(\sqrt{\lambda_{0}\mu_{0}\nu_{0}}{|000\rangle }_{ACE}\pm \sqrt{\lambda_{1}\mu_{1}\nu_{1}}{|111\rangle }_{ACE}\right)|G_{0,1}\rangle \\ \; & +\frac{1}{\sqrt{2}}\left(\sqrt{\lambda_{0}\mu_{0}\nu_{1}}{|001\rangle }_{ACE}\pm \sqrt{\lambda_{1}\mu_{1}\nu_{0}}{|110\rangle }_{ACE}\right)|G_{2,3}\rangle \\ \; & +\frac{1}{\sqrt{2}}\left(\sqrt{\lambda_{0}\mu_{1}\nu_{0}}{|010\rangle }_{ACE}\pm \sqrt{\lambda_{1}\mu_{0}\nu_{1}}{|101\rangle }_{ACE}\right)|G_{4,5}\rangle \\ \; & +\frac{1}{\sqrt{2}}\left(\sqrt{\lambda_{0}\mu_{1}\nu_{1}}{|011\rangle }_{ACE}\pm \sqrt{\lambda_{1}\mu_{0}\nu_{0}}{|100\rangle }_{ACE}\right)|G_{6,7}\rangle \;. \end{array}$$This equation shows that after measurements on BDF qubits, we gain ACE qubits in any one of the eight possible forms of entangled state. For example, if measurement gives us $|G_{0}\rangle$ then ACE qubits have state
(11)$$\frac{1}{\sqrt{2p_{0}}}\left(\sqrt{\lambda_{0}\mu_{0}\nu_{0}}{|000\rangle }_{\mathrm{ACE}}+\sqrt{\lambda_{1}\mu_{1}\nu_{1}}{|111\rangle }_{\mathrm{ACE}}\right)\;.$$Here, probability of obtaining $|G_{0}\rangle$ state is $p_{0}=\left(\lambda_{0}\mu_{0}\nu_{0}+\lambda_{1}\mu_{1}\nu_{1}\right)/2$.

Yu and Song [35] showed that any good bipartite entanglement measure $M_{A-B}$ can be extended to multipartite systems by taking bipartite partitions of them. So a tripartite entanglement quantifier can be defined as
(12)$$M_{ABC}^{+}=\frac{1}{3}\left(M_{A-BC}+M_{B-AC}+M_{C-AB}\right).$$However, $M_{ABC}^{+}$ could be non-zero for pure biseparable states. It can be avoided by using the geometric mean:(13)$$M_{ABC}^{\times }={\left(M_{A-BC}M_{B-AC}M_{C-AB}\right)}^{\frac{1}{3}}\;.$$However, we use global entanglement Q for a multiplicative definition if our entanglement quantifier *M* is tangle (the square of the concurrence) [36]. In a more general setting, this entanglement quantifier geometric mean concept was put forth in [37]. Now by considering this bi-partition for the final triqubit entangled state, we can compute the swapped entanglement in the form of concurrence as
(14)$$C_{ACE}={\left(C_{A-CE}C_{C-AE}C_{E-AC}\right)}^{\frac{1}{3}}\;,$$
where $C_{A-CE}=\sqrt{2\left(1-\mathrm{Tr}\left(\rho_{A}^{2}\right)\right)}$ and similarly we can compute $C_{C-AE}$, $C_{E-AC}$. Here, $\rho_{A}$ is the one-qubit reduced density matrix of the qubit *A*, obtained after tracing out the other qubits. The average concurrence for the final three qubits state now can be written as
(15)$$C_{ACE}^{av}=C_{AB}C_{CD}C_{EF}\;.$$It is again equal to the product of the concurrences of the initial three states.

We can compute the the negativity of triqubit state $\rho_{ACE}$ as
(16)$$N\left(\rho_{ACE}\right)={\left(N_{A-CE}N_{C-AE}N_{E-AC}\right)}^{\frac{1}{3}}\;,$$
where $N_{A-CE}=-2\sum_{i}\lambda_{i}\left(\rho_{ACE}^{T_{A}}\right)$, $\lambda_{i}\left(\rho_{ACE}^{T_{A}}\right)$ are the negative eigenvalues of $\rho_{ACE}^{T_{A}}$, partial transpose of $\rho_{ACE}$ with respect to subsystem *A* is defined as $\langle i_{A},j_{CE}\left|\rho_{ACE}^{T_{A}}\right|k_{A},l_{CE}\rangle =\langle k_{A},j_{CE}\left|\rho \right|i_{A},l_{Ce}\rangle$ and, similarly, we can define $N_{C-AE}$, $N_{E-AC}$. The average negativity of the final triqubit state can be written as
(17)$$N_{ACE}^{av}=N_{AB}N_{CD}N_{EF}\;,$$
where $N_{AB},N_{CD}$ and $N_{EF}$ are the negativities of the initial three biqubit states.

### Application of Swapped Entanglement

The final swapped entanglement between Alice and Danny’s qubit has wide range of application, however, we are interested in imposing it for teleportation of an unknown qubit state. Let, after the entanglement swapping, Alice and Danny attain the state ${|{\ddot{\Phi }}_{+}\rangle }_{AD}$ that can also be written as
(18)$${|{\ddot{\Phi }}_{+}\rangle }_{AD}=\left(a{|0\rangle }_{A}{|0\rangle }_{D}+b{|1\rangle }_{A}{|1\rangle }_{D}\right)\;,$$
where $a=\alpha_{0}^{*}\sqrt{p_{0}p_{0}^{\prime }/p_{{\tilde{\Phi }}_{+}}}$, $b=\beta_{0}^{*}\sqrt{p_{1}p_{1}^{\prime }/p_{{\tilde{\Phi }}_{+}}}$ and ${\left|a\right|}^{2}+{\left|b\right|}^{2}=1$. If $a=b=\frac{1}{\sqrt{2}}$ then Equation (Equation 18) is maximally entangled otherwise non-maximally entangled. Let Alice and Danny win a maximally entangled state after entanglement swapping and Alice wants to teleport an unknown quantum state to Danny. We denote the state that Alice wants to send as
(19)|χ〉=α|0〉+β|1〉,
where ${\left|\alpha \right|}^{2}+{\left|\beta \right|}^{2}=1$. Now the state |χ〉 can be teleported easily as described in Ref. [38]. However, if $\left|a\right|\neq \frac{1}{\sqrt{2}}$ then Alice and Danny are not sharing a maximally entangled state and in this case, we use probabilistic teleportation to transmit |χ〉. In such a situation, the receiver (Danny) cannot apply single-qubit unitary operations I,X,iY,Z on his collapsed state $\frac{\alpha a|0\rangle +\beta b|1\rangle }{\sqrt{{\left|\alpha a\right|}^{2}+{\left|\beta b\right|}^{2}}}$ to obtain |χ〉. Therefore, Danny has to prepare an ancilla qubit ${|0\rangle }_{Auxi}$ and applies $U_{0}$ unitary operation on the combined system as
(20)$$U_{0}\left(\frac{\alpha a|0\rangle +\beta b|1\rangle }{\sqrt{{\left|\alpha a\right|}^{2}+{\left|\beta b\right|}^{2}}}{|0\rangle }_{Auxi}\right)\;,$$
where
$$U_{0}=\left(\begin{array}{cccc} \frac{b}{a} & \sqrt{1-\frac{b^{2}}{a^{2}}} & 0 & 0\\ 0 & 0 & 0 & -1\\ 0 & 0 & 1 & 0\\ \sqrt{1-\frac{b^{2}}{a^{2}}} & -\frac{b}{a} & 0 & 0 \end{array}\right)\;.$$After implementation of unitary $U_{0}$, the expression (Equation 20) attains the form
(21)$$\frac{1}{\sqrt{{\left|\alpha a\right|}^{2}+{\left|\beta b\right|}^{2}}}\left(b\left(\alpha |0\rangle +\beta |1\rangle \right)|0\rangle +\alpha \sqrt{a^{2}-b^{2}}|1\rangle |1\rangle \right)\;.$$

Now Danny makes a measurement on his ancilla (right most) qubit in the computational basis {|0〉,|1〉}. If he obtains |0〉 then his state collapses to α|0〉+β|1〉, Danny further applies $I_{2\times 2}$ operation on the state obtained to reconstruct the desired state |χ〉. If the measurement of ancilla gives |1〉 then protocol fails to teleport the required state due to its probabilistic nature. Similarly, we can also explain the teleportation of an unknown qubit for other states of qubits *A* and *D* that appeared in Equation (Equation 4).

The measurement on Equation (Equation 10) gives us a three-qubit entangled state that can be any one of the eight three-qubit GHZ states. The teleportation for the three-qubit GHZ state has already been considered in Refs. [39,40].

## 3. Noisy Qubits and Entanglement Swapping

We have used so far pure quantum systems. These quantum systems are isolated from external environments which comprise a variety of disorders and noises. In reality, quantum systems interact with the environment. One of the important types of noise is called depolarizing noise or white noise. This type of noise takes a quantum state and replaces it with a completely mixed state $\frac{1}{N}I$, where *N* is the dimension of the quantum system and I is identity matrix. Let us consider a biqubit noisy state that is prepared by mixing a pure state with white noise:(22)$$\begin{array}{cc} \rho_{\alpha } & =\alpha \rho_{AB}+\left(1-\alpha \right)I_{2}⊗I_{2}/4\\ \; & =\left(\begin{array}{cccc} \frac{1-\alpha }{4}+\alpha p_{0} & 0 & 0 & \alpha \sqrt{p_{0}}\sqrt{p_{1}}\\ 0 & \frac{1-\alpha }{4} & 0 & 0\\ 0 & 0 & \frac{1-\alpha }{4} & 0\\ \alpha \sqrt{p_{0}}\sqrt{p_{1}} & 0 & 0 & \frac{1-\alpha }{4}+\alpha p_{1} \end{array}\right)\;, \end{array}$$
where $\rho_{AB}={|\phi \rangle }_{AB}\langle \phi |$ is the density operator of biqubit system AB described in Equation (Equation 1), $I_{2}$ identity matrix and parameter α called visibility of system AB. If we take $p_{0}=p_{1}=1/2$, the Equation (Equation 22) becomes an isotropic state [41] with maximally entangled $\rho_{AB}$. The isotropic states are invariant under all transformations of the form $U⊗U^{*}$, where the asterisk denotes complex conjugation in a certain basis.

We can also represent a biqubit noisy system in the Bloch form as [42]
(23)$$\begin{array}{ccc} \rho = & \frac{1}{4}I_{2}⊗I_{2}+\sum_{\mu =1}^{3}r_{\mu }\frac{\sigma_{\mu }}{\sqrt{2}}⊗\frac{I_{2}}{2}+\sum_{\nu =1}^{3}s_{\nu }\frac{I_{2}}{2}⊗\frac{\sigma_{\nu }}{\sqrt{2}}\; & +\sum_{\mu =1}^{3}\sum_{\nu =1}^{3}t_{\mu \nu }\frac{\sigma_{\mu }}{\sqrt{2}}⊗\frac{\sigma_{\nu }}{\sqrt{2}}\;, \end{array}$$
where σ represents Pauli matrices, I is identity matrix, $r_{\mu }=\mathrm{Tr}\left(\rho \frac{\sigma_{\mu }}{\sqrt{2}}⊗\frac{I_{2}}{2}\right)$ and $s_{\nu }=\mathrm{Tr}\left(\rho \frac{I_{2}}{2}⊗\frac{\sigma_{\nu }}{\sqrt{2}}\right)$ are Bloch vectors of given two qubits and $t_{\mu \nu }=\mathrm{Tr}\left(\rho \frac{\sigma_{\mu }}{\sqrt{2}}⊗\frac{\sigma_{\nu }}{\sqrt{2}}\right)$ called a correlation tensor. We can construct Bloch matrix from $\vec{r}$, $\vec{s}$ and 3×3 dimensional correlation matrix *T* as
(24)$$\tilde{T}=\left(\begin{array}{cc} c & \vec{s}\\ \vec{r} & T \end{array}\right)\;,$$
where *c* is a scalar number. The Bloch matrix form of Equation (Equation 22) contains $c=\alpha \sqrt{p_{0}}\sqrt{p_{1}}$, $\vec{r}=\vec{s}=0$ and correlation matrix
(25)$$T=\left(\begin{array}{ccc} -\alpha \sqrt{p_{0}}\sqrt{p_{1}} & 0 & 0\\ 0 & \frac{\alpha -1}{4}+\frac{1}{2}\left(\frac{1-\alpha }{4}+\alpha p_{0}\right)+\frac{1}{2}\left(\frac{1-\alpha }{4}+\alpha p_{1}\right) & \frac{1-\alpha }{8}+\frac{\alpha -1}{8}+\frac{1}{2}\left(\frac{1-\alpha }{4}+\alpha p_{0}\right)+\frac{1}{2}\left(\frac{\alpha -1}{4}-\alpha p_{1}\right)\\ 0 & \frac{1-\alpha }{8}+\frac{\alpha -1}{8}+\frac{1}{2}\left(\frac{1-\alpha }{4}+\alpha p_{0}\right)+\frac{1}{2}\left(\frac{\alpha -1}{4}-\alpha p_{1}\right) & \frac{1-\alpha }{4}+\frac{1}{2}\left(\frac{1-\alpha }{4}+\alpha p_{0}\right)+\frac{1}{2}\left(\frac{1-\alpha }{4}+\alpha p_{1}\right) \end{array}\right)\;.$$As the coherence vectors $\vec{r}$, $\vec{s}$ of the subsystems *A* and *B* have zero magnitudes that means the state is maximally mixed. According to combo separability criteria [42] if $f\left(\alpha ,p_{0}\right)={∥{\tilde{T}}_{\alpha }∥}_{KF}-1>0$ then state $\rho_{\alpha }$ is an entangled state. We plotted $f(\alpha ,p_{0})$ in Figure 2 which represents entanglement of the mixed state $\rho_{\alpha }$.

It is clear from the figure that $\rho_{\alpha }$ remains entangled when $0<p_{0}<1$ and the minimum value of α is 1/3. This state becomes maximally entangled when $p_{0}=0.5$ and α approaches 1.

As $\rho_{\alpha }$ is a *X*-form mixed-state with non-zero entries only along the diagonal and anti-diagonal so its concurrence is given by [43]
(26)$$\begin{array}{cc} C(\rho_{\alpha }) & =\max\left[0,2\left(\left|\rho_{\alpha }^{\left(14\right)}\right|-\sqrt{\rho_{\alpha }^{\left(22\right)}\rho_{\alpha }^{\left(33\right)}}\right),2\left(\left|\rho_{\alpha }^{\left(23\right)}\right|-\sqrt{\rho_{\alpha }^{\left(11\right)}\rho_{\alpha }^{\left(44\right)}}\right)\right]\\ \; & =\max\left[0,2\left(\alpha \sqrt{p_{0}p_{1}}-\frac{1-\alpha }{4}\right)\right]\;. \end{array}$$This relation also gives a lower bound for the probability that keeps the $\rho_{\alpha }$ entangled as
(27)$$\alpha >\frac{1}{\left(1+4\sqrt{p_{0}p_{1}}\right)}\;.$$

If $\rho_{AB}$ is a maximally entangled state then $p_{0}=p_{1}=\frac{1}{2}$, in this case the state remains entangled for $\alpha >\frac{1}{3}$ that we can also observe from Figure 3a.

The negativity of *X*-form state $\rho_{\alpha }$ can be computed as
(28)$$N\left(\rho_{X}\right)=-2\min\left\{0,r_{+}-\sqrt{r_{-}^{2}+{\left(\rho^{\left(14\right)}\right)}^{2}},u_{+}-\sqrt{u_{-}^{2}+{\left(\rho^{\left(23\right)}\right)}^{2}}\right\}\;,$$
where $u_{\pm }=\left(\rho^{\left(11\right)}\pm \rho^{\left(44\right)}\right)/2$, $r_{\pm }=\left(\rho^{\left(22\right)}\pm \rho^{\left(33\right)}\right)/2$. The Equation (Equation 28) can be reduced to
(29)$$N\left(\rho_{\alpha }\right)=-2\min\left\{0,\frac{1}{4}\left(1-\alpha -4\alpha \sqrt{p_{0}p_{1}}\right)\right\}\;,$$
because $\rho^{\left(23\right)}=0$ and $u_{+}-u_{-}=\rho^{\left(44\right)}$ is a positive number. For a maximally entangled state, the Equation (Equation 29) also gives $\alpha >\frac{1}{3}$ for entanglement retain and can be observed from Figure 3b.

It is clear from Figure 3 that the concurrence and negativity produce similar results in the case of biqubit noisy state.

Now we want to explore entanglement swapping between two states of the form given in Equation (Equation 22). For simplicity, we assume that the states are similar and we make standard Bell measurements in order to accomplish the entanglement swapping. Therefore our four-qubit noisy state in terms of Bell basis is
(30)$$\begin{array}{cc} \rho_{ABCD}= & \frac{\alpha^{2}}{2}\left(p_{0}{|00\rangle }_{AD}\pm p_{1}{|11\rangle }_{AD}\right)\left(p_{0}\langle 00|\pm p_{1}\langle 11|\right)|\Phi_{\pm }\rangle_{\mathrm{BC}}\langle \Phi_{\pm }|\\ \; & +\frac{\alpha^{2}p_{0}p_{1}}{2}\left({|01\rangle }_{AD}\pm {|10\rangle }_{AD}\right)\left(\langle 01|\pm \langle 10|\right)|\Psi_{\pm }\rangle_{\mathrm{BC}}\langle \Psi_{\pm }|\\ \; & +d_{AD}\left(|\Phi_{\pm }\rangle_{\mathrm{BC}}\langle \Phi_{\pm }\left|+\right|\Psi_{\pm }\rangle_{\mathrm{BC}}\langle \Psi_{\pm }|\right)\;, \end{array}$$
where
(31)$$\begin{array}{c} d_{AD}=\frac{\alpha \left(1-\alpha \right)}{8}\left(\left(p_{0}{|0\rangle }_{A}\langle 0|+p_{1}{|1\rangle }_{A}\langle 1|\right)⊗I_{D}+I_{A}⊗\left(p_{0}{|0\rangle }_{D}\langle 0|+p_{1}{|1\rangle }_{D}\langle 1|\right)\right)\\ +\frac{{\left(1-\alpha \right)}^{2}}{16}I_{AD}s\;, \end{array}$$
and I is an identity matrix. The measurement of the qubits *B* and *C* will give us one of the following four Bell states
(32)$$\begin{array}{cc} {|\Phi_{\pm }\rangle }_{BC} & =\frac{1}{\sqrt{2}}\left({|00\rangle }_{BC}\pm {|11\rangle }_{BC}\right)\;,\\ {|\Psi_{\pm }\rangle }_{BC} & =\frac{1}{\sqrt{2}}\left({|01\rangle }_{BC}\pm {|10\rangle }_{BC}\right)\;. \end{array}$$If we obtain ${|\Phi_{\pm }\rangle }_{BC}$ then the qubits *A* and *D* become entangled in the state
(33)$$\rho_{AD}^{\Phi_{\pm }}=\frac{1}{P_{\Phi }}\left(\frac{\alpha^{2}}{2}\left(p_{0}{|00\rangle }_{AD}\pm p_{1}{|11\rangle }_{AD}\right)\left(p_{0}\langle 00|\pm p_{1}\langle 11|\right)+d_{AD}\right)\;,$$
where $P_{\Phi }=\frac{\alpha^{2}}{2}\left(p_{0}^{2}+p_{1}^{2}\right)+\frac{1-\alpha^{2}}{4}$ is the probability of $|\Phi_{+}\rangle$ and $|\Phi_{-}\rangle$. If measurement gives us ${|\Psi_{\pm }\rangle }_{BC}$ then the qubits *A* and *D* make entangled state
(34)$$\rho_{AD}^{\Psi_{\pm }}=\frac{1}{P_{\Psi }}\left(\frac{\alpha^{2}p_{0}p_{1}}{2}\left({|01\rangle }_{AD}\pm {|10\rangle }_{AD}\right)\left(\langle 01|\pm \langle 10|\right)+d_{AD}\right)\;.$$Here, $P_{\Psi }=\alpha^{2}p_{0}p_{1}+\frac{1-\alpha^{2}}{4}$ is the probability of $|\Psi_{+}\rangle$ and $|\Psi_{-}\rangle$.

In order to evaluate the transferred entanglement between qubits *A* and *D*, we first compute the concurrence of all four types of density matrix $\rho_{AD}$. As all density matrices of qubits *A* and *D* in Equations (Equation 33) and (Equation 34) have *X*-form state form so, their concurrence can easily be computed by using Equation (Equation 26). The state $\rho_{AD}^{\Phi_{+}}$ and $\rho_{AD}^{\Phi_{-}}$ have the same amount of concurrence and is given by
(35)$$C\left(\rho_{AD}^{\Phi_{\pm }}\right)=\frac{1}{P_{\phi }}\max\left(0,\alpha^{2}p_{0}p_{1}-\frac{1}{8}\left(1-\alpha^{2}\right)\right)\;.$$
and similarly, the concurrence of $\rho_{AD}^{\Psi_{\pm }}$ is
(36)$$\begin{array}{c} C\left(\rho_{AD}^{\Psi_{\pm }}\right)=\frac{1}{P_{\psi }}\max\left(0,\alpha^{2}p_{0}p_{1}-\frac{1}{8}\left(1-\alpha \right)\sqrt{1+2\alpha -3\alpha^{2}+16\alpha^{2}p_{0}p_{1}}\right)\;. \end{array}$$Now the average of teleported entanglement in terms of concurrence can be computed as
(37)$$C_{av}=2P_{\Phi }C\left(\rho_{AD}^{\Phi_{+}}\right)+2P_{\Psi }C\left(\rho_{AD}^{\Psi_{+}}\right)\;.$$This average concurrence of the final states has been plotted in Figure 4 with solid lines. In addition, Figure 4 put forward that for mixed states, the product of the concurrences of the initial states (dashed line plots) is an upper bound to the average concurrence of the finally swapped entanglement (solid line plots).

We can also evaluate the transferred entanglement between qubits *A* and *D* in terms of negativity. As all density matrices of qubits *A* and *D* in Equations (Equation 33) and (Equation 34) have X-form state, hence, their negativity can easily be computed by Equation (Equation 28). The state $\rho_{AD}^{\Phi_{+}}$ and $\rho_{AD}^{\Phi_{-}}$ have the same amount of negativity and take the form
(38)$$N\left(\rho_{AD}^{\Phi_{\pm }}\right)=-\frac{2}{P_{\phi }}\min\left(0,\frac{1}{16}\left(1-\alpha^{2}\right)-\frac{1}{2}\alpha^{2}p_{0}p_{1}\right)\;.$$
and
(39)$$\begin{array}{c} N\left(\rho_{AD}^{\Psi_{\pm }}\right)=-\frac{2}{P_{\psi }}\min\left(0,\frac{1-\alpha^{2}}{16}-\sqrt{\frac{1}{4}\alpha^{4}p_{0}^{2}p_{1}^{2}+\frac{1}{64}{\left(1-\alpha \right)}^{2}\alpha^{2}\left(p_{0}-p_{1}\right){}^{2}}\right)\;. \end{array}$$The average of swapped entanglement computed by negativity can be given as
(40)$$N_{av}=2P_{\Phi }N\left(\rho_{AD}^{\Phi_{\pm }}\right)+2P_{\Psi }N\left(\rho_{AD}^{\Psi_{\pm }}\right)\;.$$The average of swapped entanglement in terms of negativity has been plotted in Figure 5 with solid lines. The product of the negativities of the initial states (dashed line plots in Figure 5) provides an upper bound to the average negativity of the final states (solid line plots). Moreover, Figure 6 represents the comparison of average concurrence and average negativity of final states. This plot shows that when $p_{0}=p_{1}=0.5$ which correspond to maximally entangled input states then concurrence and negativity overlap but for other cases, dotted and solid lines do not overlap. The rectangular block in this plot shows the zoomed view of non-overlapping parts of solid and dotted blue as well as green lines. We can see that the dotted green line deviate from zero line before the solid green line. In this particular case, we find that in the weak entanglement regimes, negativity gives better results compared to the concurrence. However, the opposite occurs when the state enters into relatively higher entanglement regime. Specifically, the weak entangled state with non-zero negativity has zero concurrence. As negativity detects all types of entangled states successfully that means negativity outperforms the concurrence.

### Teleportation Using a Noisy Quantum Channel

The four Bell states for qubits *B* and *C* are given in Equation (Equation 32). By using the similar Bell states for qubits *A* and *D*, we can define the projectors $P^{\Phi_{\pm }}={|\Phi_{\pm }\rangle }_{AD}\langle \Phi_{\pm }|$ and $P^{\Psi_{\pm }}={|\Psi_{\pm }\rangle }_{AD}\langle \Psi_{\pm }|$ associated with the measurements that Alice performs in the execution of the teleportation protocol. Equations (Equation 33) and (Equation 34) represent the density matrices of four possible outcomes after entanglement swapping of noisy entangled states. We can use these density matrices as a channel for the teleportation of an unknown qubit given in Equation (Equation 19) from Alice to Danny. Let the teleportation channel is $\rho_{AD}^{\Phi_{+}}$ and the density matrix of the qubit to be teleported is given by $\rho_{tq}=|\chi \rangle \langle \chi |$, where the subscripts *tq* means “teleportation qubit”. The initial three qubits state is given by
(41)$$\varrho_{1}=\rho_{tq}⊗\rho_{AD}^{\Phi_{+}}.$$The first two qubits (i.e., $\rho_{tq}$ and first qubit of $\rho_{AD}^{\Phi_{+}}$) of $\varrho_{1}$ are possessed by Alice and the third qubit is occupied by Danny. Alice makes a projective measurement on her two qubits. After this measurement, we attain the post-measurement state
(42)$$\tilde{\varrho_{1}}=\frac{P^{\Phi_{+}}\varrho_{1}P^{\Phi_{+}}}{\tilde{P_{1}}}\;,$$
where $\tilde{P_{1}}=\mathrm{Tr}\left[P^{\Phi_{+}}\varrho_{1}\right]$ is the probability of occurrence of state $\tilde{\varrho_{1}}$ and Tr represents trace operation. Alice then communicates her outcomes with Danny via the classical channel. The qubit possessed by Danny has the form ${\tilde{\varrho }}_{D1}={\mathrm{Tr}}_{12}\left[\tilde{\varrho_{1}}\right]$, where ${\mathrm{Tr}}_{12}$ means the partial trace of qubits 1 and 2. Due to the noisy teleportation channel, Danny has to follow a probabilistic teleportation technique to find teleported qubit $\rho_{tq}$. He prepares an auxiliary qubit $\rho_{Auxi}=|0\rangle \langle 0|$ and applies a suitable unitary operator $U_{i}$ on two qubits system as
(43)$$U_{i}\left({\tilde{\varrho }}_{D1}⊗\rho_{Auxi}\right)U_{i}^{†}\;.$$Then a measurement on Danny’s auxiliary qubit in the basis $\left\{|0\rangle \langle 0|,|0\rangle \langle 1|,|1\rangle \langle 0|,|1\rangle \langle 1|\right\}$ is completed. If |0〉〈0| occurs, we obtain qubit ${\tilde{\varrho }}_{D1}^{\prime }$ with some probability $P_{1}^{\prime }$ otherwise the teleportation fails.

The protocol ends with Danny apply a unitary operation *u* on his qubit final state as
(44)$$\rho_{tq}^{\prime }=u\frac{{\tilde{\varrho }}_{D1}^{\prime }}{P_{1}^{\prime }}u^{†}\;.$$The unitary operator *u* is one of the Pauli operators $\left\{I,\sigma_{x},\sigma_{y},\sigma_{z}\right\}$, and its choice depends not only on the measurement result of Alice but also on the quantum channel shared between Alice and Danny in the teleportation protocol.

Now we can check the efficiency of the protocol by using fidelity [44]. Since the input state $\rho_{tq}=|\chi \rangle \langle \chi |$ is pure, the fidelity can be written as
(45)$$F=\mathrm{Tr}\left[\rho_{tq}\rho_{tq}^{\prime }\right]=\langle \chi |\rho_{tq}^{\prime }|\chi \rangle \;.$$The fidelity ranges from 0 to 1 and its maximal value occurs whenever the Danny’s qubit final state $\rho_{tq}^{\prime }$ is equal to input qubit $\rho_{tq}$ and it is 0 when the two states are orthogonal.

## 4. Concluding Discussion

We have studied an entanglement swapping protocol, where Alice and Bob share a generalized Bell pair (A,B) whereas, Cara and Danny share another generalized Bell pair (C,D). When Bob and Cara, who are situated in the same laboratory perform some measurements on the pair (B,C) then initially unentangled qubits (A,D) obtain entanglement.

In the case of two couples of pure qubits, the finally entangled couple can have one of the four possible entangled states. However, if we considered three couples of entangled qubits then entanglement swapping gives us a three-qubit entangled state that can be any one of the eight possible forms of GHZ quantum states.

The important results of this study can be summarized as, if initial quantum states are maximally entangled and we make measurements in the Bell basis then average concurrence and average negativity of final states give similar results. We simply obtain the average swapped entanglement among final quantum states by taking the product of entanglement of the initially maximally entangled states. The measurement in non-maximally entangled basis during entanglement swapping degrades the swapped entanglement. The product of the entanglement of the mixed states provides an upper bound to the average swapped entanglement of final states. The entanglement quantifier concurrence fails to detect some weak entangled noisy states but negativity work well in a weak entanglement regime. On the contrary, at the higher entanglement regimes, concurrence remained more robust than the negativity for the mixed state case. Overall negativity detects all type of entangled states so negativity is more appropriate than concurrence. We also use the final output state as a channel for the teleportation of an unknown qubit from Alice to Danny. The teleportation with a pure biquibit Bell state is obvious, but we explored the probabilistic teleportation of an unknown qubit not only with non-maximally entangled channel but also with the noisy channel that we obtain after entanglement swapping.

## Figures and Tables

**Figure 1 entropy-25-00415-f001:**
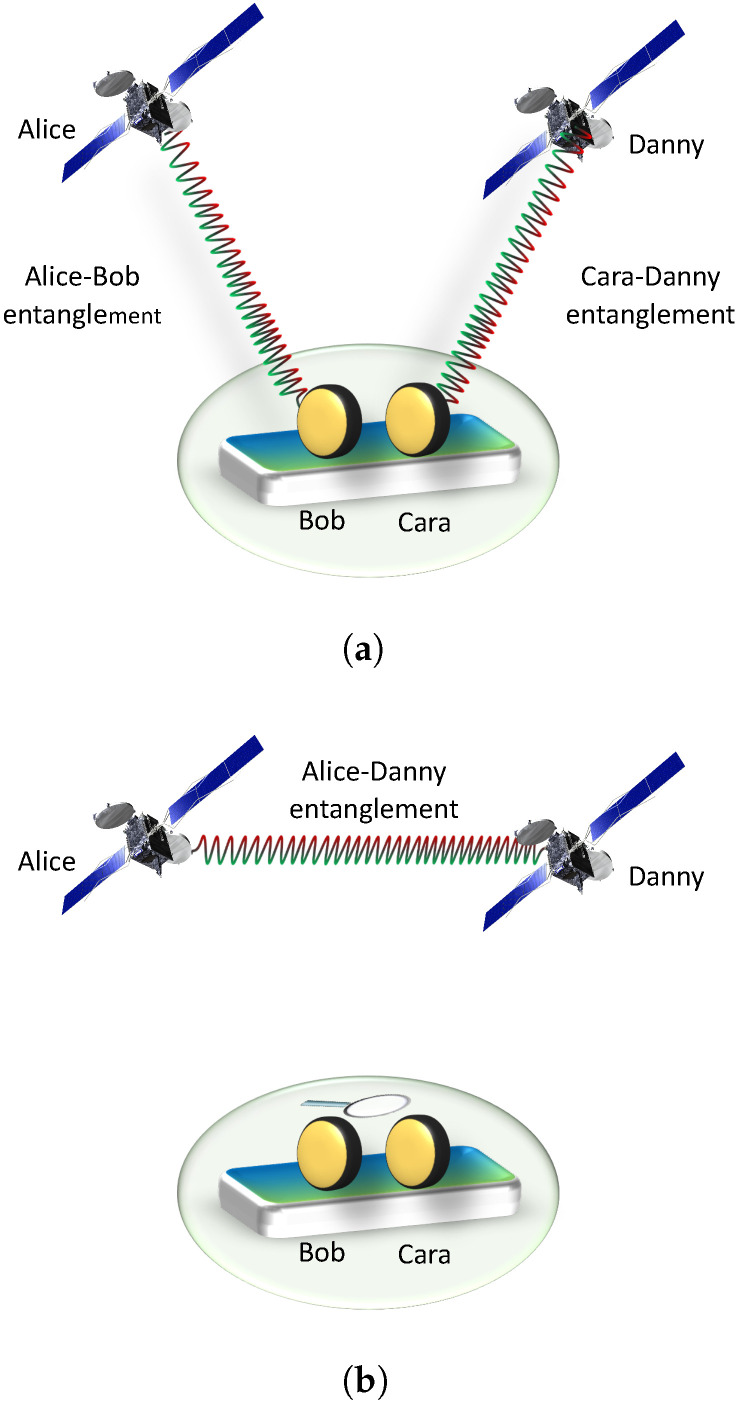
Entanglement swapping. Initially in (**a**) entangled pairs are shared between Alice and Bob, and between Cara and Danny. There is no entanglement between Alice and Danny. However, in (**b**), the measurement on Bob and Cara’s qubits project the entanglement between Alice and Danny.

**Figure 2 entropy-25-00415-f002:**
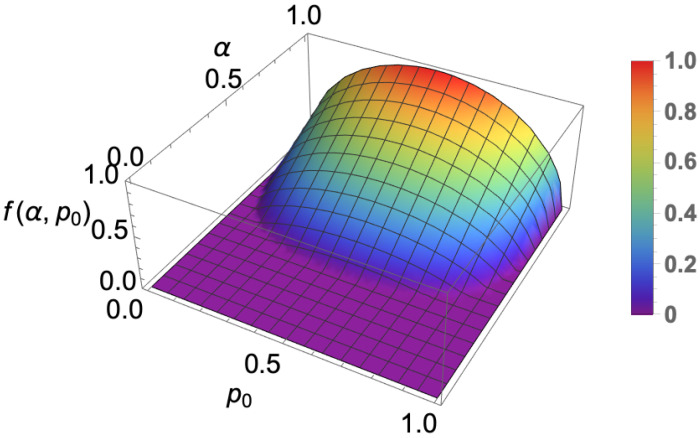
The biqubit noisy state $\rho_{\alpha }$ remains entangled for $f\left(\alpha ,p_{0}\right)={∥{\tilde{T}}_{\alpha }∥}_{KF}-1>0$.

**Figure 3 entropy-25-00415-f003:**
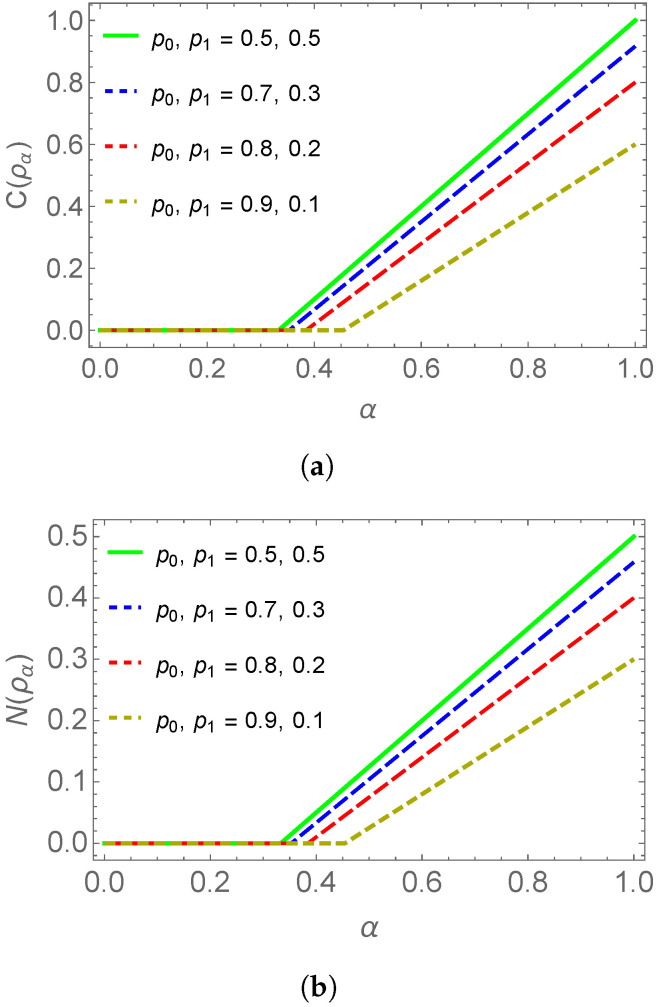
Concurrence and negativity comparison of biqubit noisy states. (**a**) represents the concurrence of biqubit state against state visibility parameter α and similar (**b**) represents the negativity of biqubit state.

**Figure 4 entropy-25-00415-f004:**
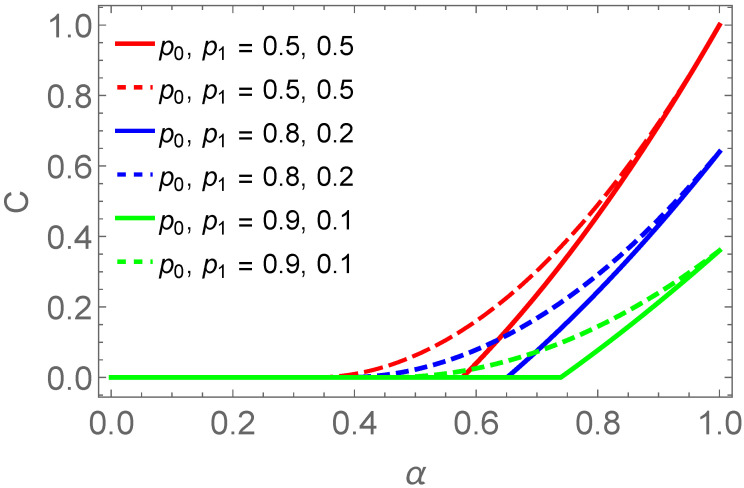
It is the comparison of the average concurrence $C_{av}$ of final states (solid lines) with the product of concurrences of input states (dashed lines) for different values of $p_{0}$ and $p_{1}$ against α.

**Figure 5 entropy-25-00415-f005:**
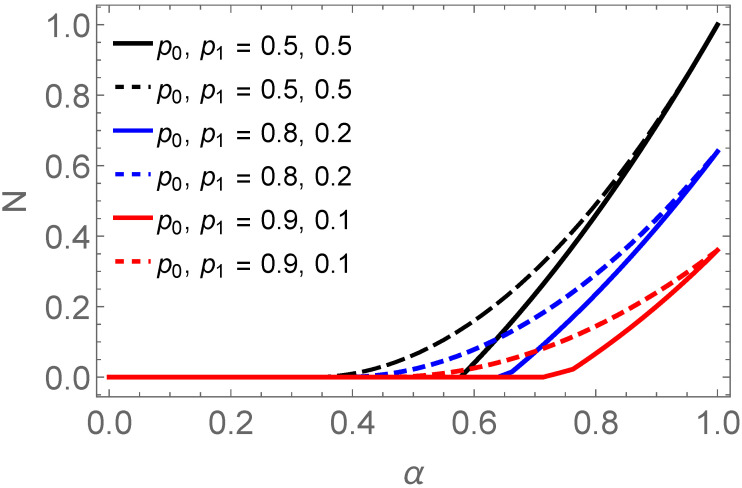
It is the comparison of the average negativity $N_{av}$ of final states (solid lines) with the product of negativities of initial states (dashed lines).

**Figure 6 entropy-25-00415-f006:**
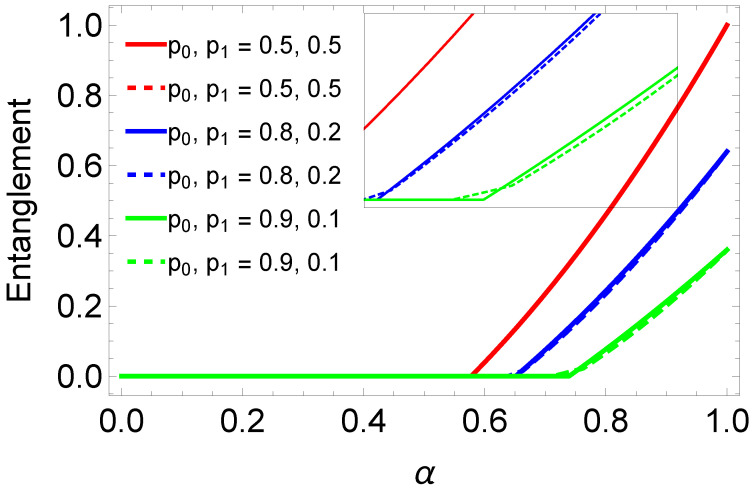
Comparison of entanglement quantifiers for swapped entanglement. Solid lines represent the average concurrence $C_{av}$ and the dashed lines represent the average negativity $N_{av}$ of the final states.

## Data Availability

The data that support the findings of this study available upon reasonable request from the authors.

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
