# Peer review of "Entanglement Swapping and Swapped Entanglement"

_entropy, 2023, doi:10.3390/e25030415_

Round 1

Reviewer 1 Report

    In this submission the authors discuss the entanglement swapping in two pairs of partially correlated qubits. Although the manuscript seems to be free of basic mistakes it looks more like an exercise quantum optics, than as a research paper. The authors just performed standard calculations and provide a simple analysis of them. Most of results included in the manuscript are quite known and even used in standard courses of quantum information. The manuscript is too long especially considering the amount of new information. The whole Sec.2 and the most of Sec. 3 contain well known results and is excessively detailed written. At the end of Sec. 3 the authors introduce an entanglement measure for  3 qubit systems, Eq.22, which is not well justified. In particular, there is no relation to the standard three-tangle, C Eltschka, J Siewert, J Phys A 47  424005 (2014). The results obtained in Sec. 4 are not surprising at all and the application to the teleportation is quite trivial.

    Technical remark: the references to the plots, p.10 and p.12 are missed.

    I cannot recommend the this manuscript for publication.

Reviewer 2 Report

This manuscript investigates the entanglement swapping of bipartite qubit states by making use of the standard protocol. The entanglement of the swapped states is quantified in terms of concurrence and negativity. Although calculation caried out in the manuscript seems sound, the results cannot be sufficient for publication. In Section 5, the authors summarize the achievements of this manuscript which are written as "the significant achievements". Unfortunately, I cannot agree with the authors on this. For instance, they write "the concurrence and the negativity give similar results". But it is not surprising since both quantifiers are entanglement monotone and they are equivalent for qubits. They also write "the concurrence is an upper bound to the negativity. Is this result physically significant? If so, the authors should elaborate that. They should clearly explain what is motivation, what is new and what is important. After all, I cannot recommend publication of this manuscript. 

Reviewer 3 Report

I reported my comments in the attached file.

Reviewer 4 Report

The manuscript entropy-2123292 presents a valuable contribution to quantum information theory. The authors explore the topic of entanglement swapping, which is a phenomenon resulting in the entanglement between two particles that were initially uncorrelated. Entanglement swapping is relevant to quantum communication and information encoding. In Sec. 2, the authors present the theoretical foundations for their research (definitions of the Schmidt decomposition, entangled states, entanglement measures, etc.). In Sec. 3, a protocol for entanglement swapping is introduced, which can be realized with either two-qubit or three-qubit states. Finally, in Sec. 4, the authors consider noisy entanglement swapping, which is done by imposing a polarization quantum channel on the two-qubit state shared by the parties A and B. For different combinations of the parameters, the figures of merit that quantify entanglement are presented graphically. Overall, the manuscript is solid and well-written. The analysis is comprehensive. I advise to accept the manuscript for publication in Entropy after a minor revision. In the revision, the authors should take into account the following points.

1. There is a problem with referring to figures in the text. Instead of the corresponding numbers, we see only question marks, which makes it difficult to follow the paper.

2. What exactly is plotted in Fig. 2? The vertical axis is not properly labeled. The description below this figure reads: "This plot represents entangled states present in the mixed state (...)", which does not explain specifically which figure of merit was plotted.

3. In Fig. 6, the authors compare the concurrence and negativity in one plot. What is the goal of this comparison? These are two different entanglement measures, so why do we need to verify the overlap between them? I think more comment is needed in this context.

4. In the case of Fig. 4, the authors should consider merging the two plots into one. It seems to me that the average concurrence is repeated in both plots. So it appears sufficient to have only the right-hand side plot. The same comment applies to Fig. 5, where negativity is presented.

Round 2

Reviewer 1 Report

I have not detected any significant improvement of the manuscript. I do not consider that the authors satisfactory replied to my main objections (extension of the manuscript, justification of the proposed measure, comparison with known entanglement measures). In addition, their comment about the novelty of the problem of entanglement swapping of mixed looks strange.

Reviewer 2 Report

The authors revised the manuscript which investigates the entanglement swapping of bipartite qubit states. They added some comments in the revised version of the manuscript. I think that it has become better than the original version. However, the authors have not answered the question given in my previous report. Is it physically important to compare the concurrence with the negativity? What does it mean that the concurrence is an upper bound to the negativity? Although the authors wrote in response, for instance, "under some specific conditions, the concurrence and negativity may coincide or may deviate a little from each other when influenced by a depolarizing noise channel", this is not an answer to my question. Hence, I can recommend publication of the present form of the manuscript.

Reviewer 3 Report

The authors followed most of the indications included in the reviewer report. However, one of the most important aspects is still unclear, i.e. the novelty of the work. The authors increased the bibliography including some works regarding entanglement-swapping protocols, but I think that these works do not allow to clarify the position of the present work concerning the literature. For example, in Ref. Kirby, Brian T., et al. "Entanglement swapping of two arbitrarily degraded entangled states." Physical Review A 94.1 (2016): 012336, a similar analysis is computed in which the entanglement swapping protocol is performed on mixed states and the relation between the concurrence of the initial and final state is analyzed. In conclusion, I am not completely satisfied with the new version of the paper and I think that the innovation of the results shown should be clarified.

Reviewer 4 Report

In this study, the authors investigate the entanglement swapping of pure and noisy systems and examine different entanglement quantifiers for quantum states. The authors find that if initial quantum states are maximally entangled and measurements are made in the Bell basis, then the average concurrence and average negativity of the final states are similar. However, measurements in non-maximally entangled bases during entanglement swapping degrade the average swapped entanglement. The study also finds that the product of the entanglement of initial mixed states provides an upper bound to the average swapped entanglement of final states. Additionally, the authors discuss how successfully the output state can be used as a channel for the teleportation of an unknown qubit.

This study is in line with the trends in quantum information theory, which focuses on understanding and manipulating entanglement in order to develop new quantum technologies such as quantum communication and quantum computing. The study's findings on the relationship between initial and final states and the impact of measurements on entanglement swapping can facalitate the development of more efficient and effective entanglement swapping protocols.

The authors have responded sufficiently to the criticism included in the review report. The changes in the presentation of the results are satisfying. In my opinion, the manuscript meets the criteria for a research article in Entropy.

Round 3

Reviewer 1 Report

Accept in  present form

Reviewer 2 Report

Although I think that the revised version of the manuscript is significantly improved, I have a new question about the inset in Figure 6, which is newly added in this version of the manuscript. The inset shows that there is a state which has non-zero negativity and zero concurrence. This is a somewhat peculiar result. I don't understand which quantifier, negativity or concurrence, is appropriate. The separable state with zero concurrence has non-zero negativity. Or the entangled state with non-zero negativity has zero concurrence. Anyway, the figure shows that either negativity or concurrence is not a good quantifier of entanglement. The authors should make this point clear.
